# Impact of Cluster B Personality Disorders in Drugs Therapeutic Community Treatment Outcomes: A Study Based on Real World Data

**DOI:** 10.3390/jcm10122572

**Published:** 2021-06-10

**Authors:** Daniel Dacosta-Sánchez, Carmen Díaz-Batanero, Fermin Fernandez-Calderon, Óscar M. Lozano

**Affiliations:** 1Department of Experimental and Clinical Psychology, University of Huelva, 21071 Huelva, Spain; daniel.daco@dpces.uhu.es (D.D.-S.); carmen.diaz@dpsi.uhu.es (C.D.-B.); fermin.fernandez@dpces.uhu.es (F.F.-C.); 2Research Center for Natural Resources, Health and Environment, University of Huelva, 21071 Huelva, Spain

**Keywords:** drug abuse, therapeutic community, outcomes, dropout, dual pathology, comorbid mental disorders

## Abstract

Background: The impact of dual pathology on treatment outcomes is unclear, with the literature reporting both favorable and unfavorable evidence. The main aim of this study was to determine how dual pathology affects treatment outcomes using real world data obtained from inpatients that began treatment in therapeutic communities. Method: The data of 2458 inpatients were used. Clinical information was obtained from electronic medical records. Reliability of diagnosis was checked and revealed a mean kappa value of 0.88. Results: Of the sample, 41.8% were discharged after achieving the therapeutic objectives. Patients diagnosed with Cluster B personality disorders were found to have a higher risk of dropping out of treatment (HR = 1.320; z = 2.61; *p* = 0.009). Conclusions: Personality traits exhibited by Cluster B patients can interfere with treatment in therapeutic communities. There is a need to develop specific interventions for these inpatient groups, which could be implemented in therapeutic communities.

## 1. Introduction

Dual pathology, understood as the temporary coexistence of two or more psychiatric disorders, one of which is problematic substance use [1], is a widely described phenomenon in the field of addictions. Some authors have pointed out that the co-occurrence of these disorders may be due to the existence of common biological, psychological, and social factors, thus increasing the risk of the joint occurrence of these disorders [2]. In general terms, patients with dual pathology have a poorer quality of life and a worse clinical course [3,4]. This could be because the interaction between the symptoms of substance use disorder and those of other mental disorders can hinder the clinical management of these patients [5]. Despite this, treatments tailored to these patients have shown successful results [1].

In the field of therapeutic communities, there is controversy regarding the impact of dual pathology on treatment retention and outcomes. While some authors have found no association between dual pathology and poor therapeutic outcomes in these contexts [6], others point to a differential impact of dual pathology depending on the type of comorbid mental disorder. For example, some authors have reported that Cluster B personality disorders are negatively associated with therapeutic success [7,8,9], while other studies did not find this association [10]. Moreover, some investigations have found that patients with depressive and anxiety symptomatology present a higher probability of leaving treatment prematurely [3,11,12], while others have found no such relationship [13,14]. Other studies have found that a diagnosis of neither anxiety nor depression, according to the diagnostic criteria of the DSM and ICD classification systems, is associated with early dropout [7,11,12,15]. Moreover, Syan et al. [16] identified a profile of patients characterized by high anxiety, depression, and PTSD scores who showed a higher risk of dropout.

Before concluding that these results are contradictory, it is necessary to consider the methodological differences between these studies. In particular, the assessment instruments used, the operationalization of the dependent variables used as outcomes, the statistical techniques employed, and the differences between participants characteristics in each study [17] could explain the heterogeneity of the results. It is also noteworthy that most of the studies conducted were observational and with sample sizes that make it difficult to control for confounding variables. It is essential to bear in mind that the estimation of regression coefficients and odds ratios associated with the different variables in these studies may be affected by the size of the groups [18,19].

In recent years, real-world data (RWD) have been used to provide complementary evidence to that obtained through clinical trials and observational studies [20]. These are not opposing methodological approaches; instead, taken together, these approaches provide a complementary way of responding to research questions that can improve health systems. Thus, the power of clinical trials to obtain scientific evidence is unquestionable, much in the same way that the possibilities of generalizing scientific evidence obtained through clinical registries should be valued [21,22]. Advances in this type of methodology have been made possible due to the development of electronic health records (EHR) and the efforts undertaken to improve the quality of such records. In this regard, some authors point out that the correct implementation of EHRs can enhance patient care quality by detecting weaknesses in the provision of services [23,24].

In the context of addictions, some authors have identified a high use of EHRs [25] in treatment centers (TCs), highlighting the potential utility of these records in research [26]. However, published studies with EHRs are relatively scarce [27,28,29,30] and even less frequent in TCs [31]. In this group of patients with high dropout rates, RWD analysis could help understand the variables associated with patient therapeutic dropout in general and the impact of comorbid mental disorders in particular. Thus, the central research question of this article was to determine if those patients diagnosed with dual pathology by the TC therapeutic teams present worse treatment outcomes, understood as higher dropout rates and lower retention time in treatment. To this end, the present study aimed to analyze RWD obtained from patients receiving treatment in a TC between the years 2015 to 2018 in order to (i) estimate the prevalence of psychopathological disorders and personality disorders diagnosed by TC professionals, (ii) analyze the relationship between the length of stay in therapeutic communities and dual pathology, and (iii) analyze whether the various psychopathological and personality disorders have an impact on dropout while controlling for sociodemographic and consumption-associated variables.

## 2. Materials and Methods

### 2.1. Design

This study employed a retrospective ex post facto design.

### 2.2. Participants

Between 1 January 2015 and 31 December 2018, a total of 2458 inpatients began treatment for a substance use disorder (SUD) in one of the 23 TCs within the Public Network of Addiction Care in Andalusia (a region of Spain with approximately 8.5 million inhabitants). Of these patients, one died during the therapeutic period and was excluded from the analysis. We excluded a further 20 patients (who began treatment in 2018 and finished in 2019) due to a lack of information on their treatment outcomes. The final sample for the analyses thus consisted of 2437 inpatients.

Patients admitted to TCs are referred from the outpatient addiction treatment centers (ATCs). In the ATCs, patients are evaluated by the clinical teams, who must issue a report on the suitability of receiving treatment in the TCs. The following criteria must be met for a patient to be admitted to a TC: (1) inadequate response to treatment or difficulty maintaining abstinence during treatment in ATCs and (2) the need for intensive treatment based on a clinical evaluation.

Assignment of patients to the various TCs is managed by a computerized system based on availability. The 23 CTs share a clinical intervention protocol that is periodically updated, including healthcare (physicians and nurses), educational, social, and psychological interventions (Arenas, 2003).

Of the participants, 84.2% were men. The mean age was 38.91 years (SD = 10.22), with women being older than men (M_fem_ = 40.25; SD_fem_ = 10.43; M_male_ = 38.66; SD_male_ = 10.16; t_(2434)_ = 2.792; *p* = 0.005, d = 0.15). Of the patients, 65.3% had primary education, 30.1% had secondary education, and 2.7% had a university education. This information was not available for 1.9% of the patients. Concerning employment status before admission to the TC, 20% were working; 61% were unemployed; 1.7% were studying, and 13% were retired. The remaining percentage performed activities at home.

Of the patients, 64.5% had been diagnosed with cocaine dependence or harmful cocaine use, 51.9% with alcohol, 31.7% with cannabis, 29% with opiates, and 10.2% with benzodiazepines. Of the patients, 53.4% had a more than one diagnosis of dependence or harmful drug use. In addition, a total of 27.1% of the patients had dual pathology, with 18.4% of patients presenting an Axis I disorder and 12.3% a personality disorder. Among those with Axis I disorders, 7.5% had neurotic disorders secondary to stressful situations and somatoform disorders, 6.3% had mood disorders, 5.5% had schizophrenia and other delusional disorders, and 0.5% had an eating disorder. Of those with personality disorders, 6.1% had been diagnosed with a Cluster B disorder, 4.3% with an unspecified personality disorder, 1.4% with a Cluster A disorder, and another 0.7% with a Cluster C disorder.

All patients were admitted to the TCs voluntarily.

### 2.3. Instruments

The information analyzed in this study was obtained from the patients’ electronic medical records. This procedure follows a standardized protocol for collecting information for all public and subsidized addiction treatment centers in Andalusia. This electronic record begins with collecting information proposed in the Treatment Demand Indicator (TDI) Standard Protocol 3.0 of the European Monitoring Centre for Drugs and Drug Addiction [32]. Subsequently, the anamnesis and clinical information relevant to the patients’ treatment during their therapeutic process is incorporated periodically. This information is stored through numerically coded variables (including patient diagnosis, sociodemographic variables, and types of treatment) and text fields (including therapeutic objectives, patient progress, and family history). All clinicians in public and subsidized treatment centers have received training in the use of the electronic registry.

The patient data used in the present study were as follows:

*Sociodemographic information*. Age, gender, educational level, and employment status were recorded according to the protocol of the “Indicator of Admissions to Treatment for Psychoactive Substance Use” of the Spanish Observatory on Drugs and Drug Addiction [33].

*Diagnoses of Substance Use Disorders and other mental disorders.* Mental disorders were diagnosed using the Spanish version of the Classification of Mental and Behavioral Disorders, Tenth Revision [34]. To test reliability of diagnosis, in this study, we calculated inter-observer agreement of the diagnosis made by different clinicians. We used both the diagnoses made by the ATC clinicians (who, when requesting the admission of patients to the TCs, must prepare a report that includes all the clinical information relevant to their treatment) and the diagnoses of the TC clinicians, who must produce a clinical report when the patients are discharged from the TC. Diagnoses related to SUDs showed percentages of agreement ranging from 70%–89.2%, except for the diagnosis of MDMA (50%). The mean kappa value estimated for the different substances was 0.8. The diagnoses of comorbid mental disorders showed percentages of agreement ranging from 70%–97.5%, with a mean kappa value of 0.88.

In this study, each patient was assigned the diagnosis provided by the TC clinicians. These were made at the end of treatment, and therefore, it is less likely that the diagnosis was affected by substance use, which occurs more frequently in the outpatient setting.

*Therapeutic discharges vs. dropout.* In the present study, therapeutic discharge was the code assigned to those patients who, according to clinical criteria, had achieved the established therapeutic objectives. The criteria established by the intervention protocol [35] for the therapeutic discharge of patients is determined by (i) reduction and abstinence from drug use; (ii) acquisition of personal and social resources (e.g., attitudes, impulse control, and social skills) to cope with high-risk drug use situations; and (iii) acquisition of skills and competencies to reduce the severity of problems associated with employment, along with legal, economic, and environment issues. Once the TC treatment was completed, these patients were referred to the ATCs to continue with their treatment.

In contrast, those patients who dropped out of treatment before achieving the therapeutic objectives and those expelled from the TCs (i.e., patients who consumed drugs during their stay in TCs or exhibited aggressive behavior) were coded as dropouts. That is, a patient was considered to have dropped out of treatment when premature discontinuation of treatment occurred [36]. According to the electronic medical records, none of the patients who dropped out of treatment attended the treatment centers of the Public Network of Addiction Care in Andalusia for three months after withdrawing from treatment in the TCs.

*Treatment time* was coded according to the number of days between the patient’s admission to TC until the end of TC treatment (for any of the reasons previously indicated).

### 2.4. Procedure

In 2014, the patients’ electronic medical record was created through the Information System of the Andalusian Drug Program (Andalusia, Spain) and included all the patients’ clinical information required for their treatment. The present study used the data from 1 January 2015 to give clinicians an extensive training period in the registry and reduce coding errors.

The researchers requested the database from the General Secretariat of Social Services of the Department of Equality and Social Policies of the Junta de Andalucía (Spain). The registration of the data and storage of the information complied with the General Health Law of 25 April 1986 (Spain) and the 41/2002 Law of 14 November on patient autonomy and rights and obligations regarding clinical information and documentation along with the Organic Law 3/2018 of 5 December 2018, on the protection of personal data and guarantee of digital rights, adapted to European regulations.

The databases were sent to the project’s principal investigator in an anonymous format, as it was impossible to identify the patients. Once the database had been obtained, information on the procedure used to obtain this database was sent to the Research Ethics Committee of the Andalusian regional Ministry of Health, which certified the appropriate application of the procedure and compliance with the ethical principles for handling the information.

### 2.5. Analysis

Univariate and bivariate statistics were used to describe the sample. The associations between the different variables were analyzed using Pearson’s Chi-squared and *t*-student tests. Given the study’s sample size, effect sizes were calculated using Cramer’s V or Cohen’s d.

The risk of treatment abandonment was examined by applying Cox regression analysis. The independent variables introduced in the model were sociodemographic (variables with more than two categories were coded as dummy variables), the SUD diagnoses for each substance, and the diagnoses of other mental disorders. Time in treatment was used as the time variable and the dependent variable was the type of discharge, using a value of “1” to code patients who had dropped out of treatment and a value of “0” for patients with therapeutic discharge.

All analyses were conducted using the STATA V.14 statistical package.

## 3. Results

### 3.1. Sociodemographic Characteristics and Consumption Profile According to Comorbid Mental Disorders

Dual pathology was associated with gender since a higher percentage of women had dual pathology (40.5%) than men (24.6%), these differences being statistically significant (*χ*^2^ = 41.728; *p* < 0.001). It was also observed that patients with dual pathology were older than those patients without (M_DP_ = 39.69; SD_DP_ = 10.17 vs. M_NDP_ = 38.62; SD_NDP_ = 10.22; t_(2434)_ = 2.30; *p* = 0.021; d = 0.10). Concerning employment status, those patients with dual pathology included a higher percentage of pensioners (25.5%) than the group of patients without dual pathology (8.4%), which was statistically significant (*χ*^2^ = 124.57; *p* < 0.001). No association was found between dual pathology and educational level.

The analysis regarding dual pathology and drug use disorders showed that among the patients with psychopathological disorders, there was a statistically significant association with alcohol-related disorders (50.0% of those with dual pathology showed alcohol dependence compared with 40.2% patients without dual psychopathology, *χ*^2^ = 15.89; *p* < 0.001). Specifically, patients with alcohol dependence showed higher rates of mood disorders (8.6% vs. 3.8%; *χ*^2^ = 23.451; *p* < 0.001) and of anxiety, dissociative, stress-related, somatoform and other non-psychotic mental disorders (8.9% vs. 5.9%; *χ*^2^ = 8.165; *p* = 0.004). Patients with cannabis dependence showed higher rates of schizophrenia, schizotypal, delusional, and other non-mood psychotic disorders (10.1% vs. 3.3%; *χ*^2^ = 47.096; *p* < 0.001).

In contrast, we observed a lower prevalence of opioid dependence (18.8% of patients with psychopathology compared with 30.0% of patients without psychopathology, *χ*^2^ = 23.42; *p* < 0.001). Compared with patients without drug dependence, patients with dependence on this drug showed lower rates of mood disorders (4.2% vs. 7.2%; *χ*^2^ = 7.194; *p* = 0.007) and anxiety, dissociative, stress-related, somatoform, and other nonpsychotic mental disorders (8.4% vs. 5.2%; *χ*^2^ = 7.136; *p* = 0.008). It was also observed that patients with cocaine dependence had lower rates of dual pathology. (48.7% of patients with psychopathology and 63.7% of patients without psychopathology, *χ*^2^ = 36.26; *p* < 0.001). Specifically, these patients presented lower rates of mood disorders (4.5% vs. 9.7%; *χ*^2^ = 26.184; *p* < 0.001) along with anxiety, dissociative, stress-related, somatoform, and other non-psychotic mental disorders (10.3% vs. 5.9%; *χ*^2^ = 15.543; *p* < 0.001).

No association was observed between any type of drug dependence and harmful drug use, and personality disorders.

### 3.2. Association between Patient Characteristics and Type of Discharge

Of the patients, 41.8% were discharged after achieving the therapeutic objectives. While the type of discharge was not associated with gender, an association was found with age. Specifically, patients who successfully completed treatment were older than those who dropped out (M_therap_ = 40.35, SD_therap_ = 10.05 vs. M_No therap_ = 37.89, SD_No therap =_ 10.21; t_(2434)_ = 5.907; *p* < 0.001; d = 0.24).

Drug use disorder was also associated with the type of patient discharge. As shown in Table 1, patients diagnosed with harmful alcohol use or alcohol dependence presented more therapeutic discharges. In contrast, patients with opiate, benzodiazepine, and cannabis use disorders dropped out of treatment to a greater extent.

Regarding psychopathological disorders (Table 2), there was an association between the type of discharge and patients diagnosed with psychotic disorders. Among those with a personality disorder, patients with a Cluster B diagnosis and those with unspecified personality disorders had higher dropout rates than the rest of the patients.

### 3.3. Survival Analysis

Cox regression analysis revealed that age (HR = 0.983; z = −5.54; *p* < 0.001) and having secondary education (HR = 0.808 z = −3.58; *p* < 0.001) were associated with a lower probability of dropping out of treatment. In contrast, being a pensioner (HR = 1.352; z = 3.58; *p* < 0.001) increased the probability of dropping out of treatment. Among those patients with a drug use diagnosis, those with harmful use or dependence on opiates (HR = 1.194; z = 2.86; *p* = 0.004) or cannabis (HR = 1.275; z = 4.04; *p* < 0.001) were more likely to drop out of treatment. None of the Axis I disorder diagnoses were associated with an increased risk of dropping out of treatment. In contrast, patients diagnosed with Cluster B personality disorders (HR = 1.320; z = 2.61; *p* = 0.009) were found to have a higher risk of dropping out of treatment (Figure 1).

Analysis of the interactions between dependence on different drugs and other comorbid mental disorders did not reveal any statistically significant coefficient.

## 4. Discussion

The present study aimed to analyze the impact of comorbid mental disorders on the length of stay in therapeutic communities and success in patients undergoing treatment in TCs. Unlike previous studies, this study made use of EHRs. These data complement the evidence obtained through observational studies and clinical trials. This type of registry has also allowed access to a larger sample, and thus, the present study is among those TC studies with the largest sample sizes. Consequently, it has been possible to provide more precise statistical estimates [19]. Moreover, unlike another study that used EHR data [31], we analyzed the impact of mental disorders diagnosed according to ICD-10 criteria, one of the most widely used nosological classifications in the clinical context.

In terms of prevalence, this study has found a lower percentage of patients with mood disorders, anxiety disorders, and personality disorders than other studies [3,9,11]. This discrepancy could be because the patients were diagnosed during their treatment, and at that time, they had been abstinent for some time and were therefore likely to have shown fewer drug-induced comorbid symptoms. The latter might also explain the similarity found between the prevalence data reported here and the number of independent mood disorders and independent anxiety disorders observed in the study by Vergara-Moragues et al. [7] and Fernández Calderón et al. [37]. However, a lower prevalence of personality disorders was detected in this study than in other studies [3,7,8].

Moreover, more than half of the patients did not achieve their therapeutic objectives. This number is slightly higher than that observed by Baker et al. [31] and is in line with other results found in observational studies [3,14]. These dropout rates highlight the therapeutic complexity of TC treatment for patients with SUD. Moreover, it is essential to note that treatment abandonment harms patients’ health [38], as well as increases healthcare system costs that could be avoided if patients complete their treatment [39]. Therefore, there is a need for further clinical studies to determine which variables contribute to patients’ therapeutic success.

This study has shown that patients with opioid and cannabis use disorder, along with patients diagnosed with Cluster B personality disorders, are those most likely to drop out of treatment without achieving their therapeutic goals. The evidence for the impact of drugs of abuse on therapeutic outcomes is mixed [10,13,15,36,40], and the impact of poly-drug use should probably be taken into account [41]. Thus, it is difficult to claim that a particular drug use disorder (e.g., cocaine, opiates) is associated with poor treatment outcomes, and any differences found between studies are most likely the result of variations in the characteristics of the samples.

Regarding the higher probability of dropout and lower retention rates observed in Cluster B PD patients, our findings converge with those reported by other authors [7,8,9]. The underlying causes of this higher probability of dropout are suggested at different levels of analysis. At the neuropsychological level, although there is no unanimity, some authors have found that patients with comorbid SUD and Cluster B PD show more significant impairments in decision-making and impulsivity compared with SUD patients without dual pathology [42], with such impairments also being detected at a neuroanatomical level [43]. This cognitive impairment has been shown to predict worse therapeutic outcomes [44]. Congruent with neuropsychological findings, in a phenotypical level, Cluster B PD patients appear to have problematic interpersonal relationships [45]. This relationship style is likely to interfere with the therapeutic alliance since Olesek et al. [46] found that high levels of Cluster B traits represented a barrier to forming quality therapeutic alliances in TCs. Similarly, Levy et al. [47] found that cognitive impairments associated with decision-making and impulsivity interfered with the therapeutic alliance. Thus, it could be hypothesized that, at least partially, the higher likelihood of treatment abandonment of Cluster B PD patients than other patients with SUD could be due to impairments that affect decision making and impulsivity.

In addition to the above, it should be borne in mind that patients with dual pathology usually present greater severity and complexity of symptoms. Therefore, the therapeutic approach to these patients requires professionals specialized in dual pathology [48], who can plan a comprehensive intervention for mental health problems adapted to the needs of each patient [49]. However, addiction centers do not always have the necessary resources for the treatment of these patients [50,51], which can generate dissatisfaction with the therapy and increase the risk of drop-out [52]. The present study analyzed potentially relevant variables for determining therapeutic success in a large sample of patients treated for dual pathology in TCs. However, it is important to note certain limitations of this study. While the information stored in these electronic records has high ecological validity (since health professionals record this information for patient treatment), the researchers have very little control over the information collected. For this reason, it can only be assumed that the experience of health professionals and the use of standardized protocols ensure valid data collection. In this study, it has been possible to evaluate the reliability of clinical diagnoses by cross-checking the information recorded by the clinicians of outpatient treatment centers with the diagnoses provided by the TC clinicians. As we have seen, the level of agreement between experts has been high. However, the prevalence rates of comorbid mental disorders were found to be lower than those observed in other studies. It is worth questioning, therefore, whether recording diagnoses through the patients’ clinical history could lead to an underestimation of the diagnoses of comorbid mental disorders.

A final limitation concerns the gender of the patients since 84.2% of the patients in this study were male. Consequently, the study results should be interpreted with caution when attempting to generalize the findings to women. This limitation, however, is found in most studies conducted with SUD patients since men generally account for a higher percentage of these patients. Therefore, it would be of interest to carry out specific studies on women to confirm the stability of the findings reported here.

## Figures and Tables

**Figure 1 jcm-10-02572-f001:**
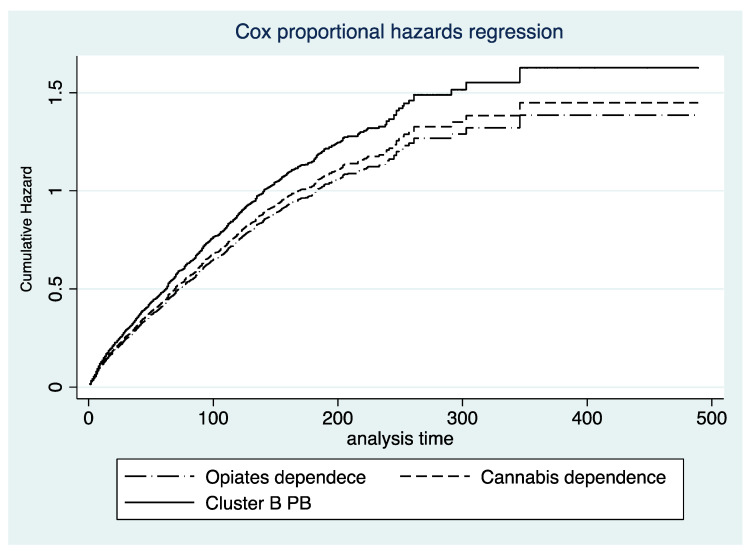
Treatment dropout risk according to patient characteristics.

**Table 1 jcm-10-02572-t001:** Sociodemographic and substance use disorders according to type of discharge.

	N (%)	Dropout (n = 1419)	Therapeutic Discharge (n = 1018)	Statistic (*χ*^2^ or Student t)	Significance	Effect Size (Cramer’s V or Cohen’s d)
Male	2051 (84.2)	84.5	83.8	0.214	0.644	0.009
Age [Mean (SD)]	2436	37.88 (10.21)	40.35 (10.05)	−5.907	0.000	0.24
**Education**						
Primary studies	1591 (65.3)	981 (69.1)	610 (59.9)	22.194	0.000	0.095
Secondary studies	799 (32.8)	413 (29.1)	386 (37.9)	20.889	0.000	0.093
**Employment status**						
Employed	487 (20.0)	259 (18.3)	228 (22.4)	6.368	0.012	0.051
Unemployed	1578 (64.8)	927 (65.3)	651 (63.9)	0.818	0.366	0.018
Pensioners	318 (13)	202 (14.2)	116 (11.4)	4.215	0.040	0.042
Students	41 (1.7)	25 (1.8)	16 (1.6)	0.129	0.719	0.007
**Harmful drug use/dependence (according to ICD-10)**						
Alcohol	1265 (51.9)	703 (49.5)	562 (55.2)	7.618	0.006	0.056
Cocaine	1573 (64.5)	934 (65.8)	639 (62.8)	2.411	0.120	0.031
Opiates	706 (29.0)	457 (32.2)	249 (24.5)	17.284	0.000	0.084
Cannabis	773 (31.7)	519 (36.6)	254 (25.0)	36.981	0.000	0.123
Benzodiazepines	249 (10.2)	160 (11.3)	89 (8.7)	4.146	0.042	0.041

**Table 2 jcm-10-02572-t002:** Comorbid mental disorder disorders according to type of discharge.

	N (%)	Dropout (n = 1419)	Therapeutic Discharge (n = 1018)	Statistic (*χ*^2^ or Student t)	Significance	Effect Size (Cramer’s V)
Dual pathology patients	660 (27.1)	415 (29.2)	245 (24.1)	8.051	0.005	0.057
Some Axis I Disorders	448 (18.4)	268 (18.9)	180 (17.7)	0.573	0.449	0.015
Mood disorders	154 (6.3)	86 (6.1)	68 (6.7)	0.384	0.536	0.013
Anxiety, dissociative, stress-related, somatoform, and other nonpsychotic mental disorders	182 (7.5)	104 (7.3)	78 (7.7)	0.095	0.758	0.006
Schizophrenia, schizotypal, delusional, and other non-mood psychotic disorders	133 (5.5)	89 (6.3)	44 (4.3)	4.368	0.037	0.042
Disorders of adult personality and behavior	299 (12.3)	202 (14.2)	97 (9.5)	12.200	0.000	0.071
Cluster A personality disorders	33 (1.4)	22 (1.6)	11 (1.1)	0.980	0.322	0.02
Cluster B personality disorders	149 (6.1)	104 (7.3)	45 (4.4)	8.736	0.003	0.06
Cluster C personality disorders	17 (0.7)	8 (0.6)	9 (0.9)	0.878	0.349	0.019
Unspecified personality disorder	106 (4.3)	73 (5.1)	33 (3.2)	5.159	0.023	0.046

## Data Availability

Database should be requested to General Secretariat of Social Services of the Department of Equality and Social Policies of the Junta de Andalucía (Spain).

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
