# Peer review of "Impact of Cluster B Personality Disorders in Drugs Therapeutic Community Treatment Outcomes: A Study Based on Real World Data"

_jcm, 2021, doi:10.3390/jcm10122572_

Round 1
Reviewer 1 Report
Based on a sample of 2,437 inpatients treated for substance abuse, the authors examined the predictive value of co-morbid cluster B personality disorders. Patients were referred from outpatient centers when they either did not respond to therapy, or needed more intensive medical treatment. The vast majority of patients were male (84.2%). According to ICD10, half of the patients (53.4%) had received more than one diagnosis of substance dependence. The distribution of ICD10 psychiatric disorders is unfortunately missing. The data analyses relied exclusively on information that the authors extracted from the electronic patient records. Outcome criteria were response to treatment, premature drop-out, and treatment retention.
The authors found a responder rate of 41.8% along with a higher drop-out risk for patients with cluster B personality disorders (p=0.009). And the authors concluded “specific interventions might be needed for this patient group in therapeutic communities”.
Comments
The manuscript addresses a topic of major clinical relevance, which was investigated by means of a pretty large sample.
However, the readability of the manuscript leaves much to be desired. Central terms of the analysis are insufficiently defined: (1) what exactly is meant with dual pathology? (2) What understand the authors under treatment retention -- regular discharge? With or without transition to another institution? Any censoring in the Cox model? (3) Treatment response differs from one patient to the other – is there a quantitative score?
Without precise, reproducible definitions, cross-comparisons with other studies is not possible and the scientific value of the study greatly reduced.
The same applies to the distribution of psychiatric diagnoses in the sample under investigation. Without this information, the potential reader cannot understand which population was actually studied here.
The manuscript would benefit greatly from a clearer structure that is comprehensible to non-specialist readers: clearly defined research question, comprehensive description of data used, model, results.
Major revision
Author Response
(1) what exactly is meant with dual pathology?
To answer the reviewer's question, we considered it appropriate to include, in the first paragraph of the Introduction, its definition as understood by the European Monitoring Center for Drugs and Drug Addiction (Torrens, Mestre-Pintó & Domingo-Salvany 2015):
“Dual pathology, understood as the temporary coexistence of two or more psychiatric disorders, one of which is problematic substance use (Torrens, Mestre-Pintó & Domingo-Salvany, 2015), is a widely described phenomenon in the field of addictions.”
(2) What understand the authors under treatment retention -- regular discharge? With or without transition to another institution?
We thank the reviewer for this comment, since the concept of "treatment retention" can be confusing. The dependent variables used in the study were "treatment time", defined as the number of days that patients were in treatment in the therapeutic community; and "therapeutic discharge/dropout", understood as those patients who achieved the therapeutic objectives.
We have proceeded to revise the MS, introducing the appropriate changes to avoid confusion for readers.
We are also grateful for your comments on the transition to another institution. We have clarified this aspect in the text:
“…That is, a patient was considered to have dropped out of treatment when premature discontinuation of treatment occurred (Brorson, Arnevik, Rand-Hendriksen, & Duckert, 2013). According to the electronic medical records, none of the patients who dropped out of treatment attended the treatment centers of the Public Network of Addiction Care in Andalusia for three months after withdrawing from treatment in the TCs.”
Any censoring in the Cox model?
We thank the reviewer for the comment on possible censored data.
As indicated in the "participants" section, 20 patients were excluded who started treatment during the study period 01/01/2015 to 12/31/2018 but completed treatment outside the study period (in 2019). One patient who died was also excluded. These 21 patients could have been considered censored data in the Cox regression model.
However, since the sample size allows estimation of the statistics with sufficient power, we chose to exclude them from the analyses.
Of the 2437 patients in the study, we have all the information on time in treatment and type of discharge/dropout, so there are no patients with censored data.
(3) Treatment response differs from one patient to the other – is there a quantitative score? Without precise, reproducible definitions, cross-comparisons with other studies is not possible and the scientific value of the study greatly reduced.
We thank the reviewer for this question.
The criteria by which patients receive therapeutic discharge are the same for all patients and are governed by the TC intervention protocol. This protocol is common to all the centers of the Public Network of Addiction Care in Andalusia.
Also, the objectives to be achieved in the TC of Andalusia are comparable to those of TCs in other countries; and, in general, they follow the guidelines indicated on the website of the National Institute on Drug Abuse (https://www.drugabuse.gov/publications/research-reports/therapeutic-communities/what-are-therapeutic-communities). Moreover, the criterion followed to determine treatment dropout is equivalent to that used by other authors (i.e., Andersson et al., 2018) and is compatible with that set out in the review study on treatment dropout by Brorson et al. (2013).
In order to avoid confusion for readers, we have proceeded to incorporate the criteria underlying therapeutic discharge in the SM.
“Therapeutic discharges vs dropout. In the present study, therapeutic discharge was the code assigned to those patients who, according to clinical criteria, had achieved the established therapeutic objectives. The criteria established by the intervention protocol (Arenas, 2003) for the therapeutic discharge of patients is determined by: i) reduction and abstinence from drug use; ii) acquisition of personal and social resources (e.g., attitudes, impulse control, and social skills) to cope with high-risk drug use situations; and iii) acquisition of skills and competencies to reduce the severity of problems associated with employment, along with legal, economic, and environment issues. Once the TC treatment was completed, these patients were referred to the ATCs to continue with their treatment.
In contrast, those patients who dropped out of treatment before achieving the therapeutic objectives and those expelled from the TCs (i.e., patients who consumed drugs during their stay in TCs or exhibited aggressive behavior) were coded as dropouts. That is, a patient was considered to have dropped out of treatment when premature discontinuation of treatment occurred (Brorson, Arnevik, Rand-Hendriksen, & Duckert, 2013). According to the electronic medical records, none of the patients who dropped out of treatment attended the treatment centers of the Public Network of Addiction Care in Andalusia for three months after withdrawing from treatment in the TCs.”
The coding assigned was "0" for therapeutic discharge and "1" for treatment abandonment. These values have been indicated in the "Analysis" section.
“…Time in treatment was used as the time variable and the dependent variable was the type of discharge, using a value of "1" to code patients who had dropped out of treatment and a value of "0" for patients with therapeutic discharge.”
The same applies to the distribution of psychiatric diagnoses in the sample under investigation. Without this information, the potential reader cannot understand which population was actually studied here.
We agree with the reviewer on the need to provide this information. The distribution of psychiatric diagnoses was included in the Results section. However, readers may find it more useful if it is presented in the description of the sample. Therefore, we have moved the information to this section.
If the reviewer is referring to any aspect not covered by this modification, please clarify.
The manuscript would benefit greatly from a clearer structure that is comprehensible to non-specialist readers: clearly defined research question, comprehensive description of data used, model, results.
We are grateful for your comments, as we believe that these suggestions may help readers to better understand the MS.
We have made the research question more explicit by including it at the end of the Introduction. The text now reads as follows:
“In the context of addictions, some authors have identified a high use of EHRs (Spivak et al., 2021) in treatment centers (TCs), highlighting the potential utility of these records in research (Marsch et al., 2020). However, published studies with EHRs are relatively scarce (Friesen et al., 2020; Krawczyk et al., 2017; Loree et al., 2019; Nasir et al., 2021) and even less frequent in TCs (Baker et al., 2020). In this group of patients with high dropout rates, RWD analysis could help understand the variables associated with patient therapeutic dropout in general and the impact of comorbid mental disorders in particular. Thus, the central research question of this article was to determine if those patients diagnosed with dual pathology by the TC therapeutic teams present worse treatment outcomes, understood as higher dropout rates and lower retention time in treatment. To this end, the present study aimed to analyze RWD obtained from patients receiving treatment in a TC between the years 2015 to 2018 in order to: i) estimate the prevalence of psychopathological disorders and personality disorders diagnosed by TC professionals; ii) analyze the relationship between the length of stay in therapeutic communities and dual pathology; iii) analyze whether the various psychopathological and personality disorders have an impact on dropout while controlling for sociodemographic and consumption-associated variables.”
Moreover, in the section on instruments, we have clarified which variables were used in the MS and the origin and coding procedure followed. Given the reorganization, we would ask the reviewer to check this in the MS.
Also, the variables that comprise the model have been specified in the analysis section. We believe that this allows the results to be better understood. The wording is as follows:
“The risk of treatment abandonment was examined by applying Cox regression analysis. The independent variables introduced in the model were sociodemographic (variables with more than two categories were coded as dummy variables), the SUD diagnoses for each substance, and the diagnoses of other mental disorders.”
We thank the reviewer for their efforts to help us improve the MS. If you feel that we should clarify anything else, please let us know.

Reviewer 2 Report
Thank you for the opportunity to review this manuscript.
The authors present an EHR-based analysis describing the risk of early treatment cessation in patients with dual pathology and abuse of specific substances. The HER-based approach is novel and provides a large set of study participants.
I have only minor comments for this manuscript:
Abstract:
Although the manuscript itself is well-written and has no issues with English proficiency, the language in the abstract is very difficult to follow and should be re-written. EG: “The main of this study is to know how impact the dual pathology in outcomes treatment, using real world data obtained from inpatients began treatment in therapeutic communities.”
Introduction:
I would appreciate slightly more elaboration as to why dual pathology might increase the likelihood of leaving treatment early. This would form a nice first paragraph for the introduction, I think.
The current first introductory paragraph is somewhat repetitive and could be made shorter and more straightforward:
Studies such as that of Syan et al. (2020) found that neither anxiety nor depression symptomatology predicted treatment dropout…. Other studies have found that a diagnosis of neither anxiety nor depression, according to the diagnostic criteria of the DSM and ICD classification systems, is associated with early dropout.
Methods/Approach:
Authors test for the impact of specific dual pathologies, and substance abuse type (eg, opioid, cocaine) on likelihood of early treatment cessation; however, I do not see any test for interactions between these. Is it possible that dual pathology also predicts the type of substance abused? EG: Do individuals with personality disorders preferentially abuse a certain type of substance? It would be edifying to include some investigation of this, and to include interaction tests here. This may also go some way to elucidating why some dual pathologies lead to increased likelihood of dropping out of treatment.
Although it is possibly outside the scope of this study, authors could consider a larger, hypothesis-free approach to probe a range of explanations for early treatment leaving; for example, would it be possible to perform a phenome-wide association study (PheWAS) using treatment outcome as the test variable? This may provide interesting additional insights beyond the analysis described here.
Discussion:
Authors observe lower rates than expected of mood, anxiety and personality disorders within this sample. It seems likely to me that this may be due to the EHR-based approach; patients may have been diagnosed previously, at other centres, or have their diagnoses listed in clinician notes rather than in ICD codes. This limitation of using EHR should be discussed by authors.
It would also be interesting to discuss the potential reasons that underly the relationship between these factors (substance abuse and dual pathologies) and treatment outcome in more detail.
Author Response
Abstract:
Although the manuscript itself is well-written and has no issues with English proficiency, the language in the abstract is very difficult to follow and should be re-written. EG: “The main of this study is to know how impact the dual pathology in outcomes treatment, using real world data obtained from inpatients began treatment in therapeutic communities.”
Thank you for this suggestion. The abstract has now been reviewed by a native speaking scientific editor to ensure that it is easier to follow.
Introduction:
I would appreciate slightly more elaboration as to why dual pathology might increase the likelihood of leaving treatment early. This would form a nice first paragraph for the introduction, I think.
We appreciate your suggestion. We have introduced a first paragraph in which a brief explanation is given as to why the risk of abandoning treatment might increase.
“Dual pathology, understood as the temporary coexistence of two or more psychiatric disorders, one of which is problematic substance use (Torrens, Mestre-Pintó & Domingo-Salvany, 2015), is a widely described phenomenon in the field of addictions. Some authors have pointed out that the co-occurrence of these disorders may occur due to the existence of common biological, psychological, and social factors, thus increasing the risk of the joint occurrence of these disorders (Deady, Tesson & Brady, 2013). In general terms, patients with dual pathology have a poorer quality of life and a worse clinical course (Daigre et al., 2017; Lozano, Rojas & Fernández-Calderón, 2017). This could be because the interaction between the symptoms of substance use disorder and those of other mental disorders can hinder the clinical management of these patients (Morisano, Babor & Robaina, 2014). Despite this, treatments tailored to these patients have shown successful results (Torrens, Mestre-Pintó & Domingo-Salvany, 2015).”
The current first introductory paragraph is somewhat repetitive and could be made shorter and more straightforward:
Studies such as that of Syan et al. (2020) found that neither anxiety nor depression symptomatology predicted treatment dropout…. Other studies have found that a diagnosis of neither anxiety nor depression, according to the diagnostic criteria of the DSM and ICD classification systems, is associated with early dropout.
Following your suggestions, we have now modified this paragraph to reduce repetition and improve the readability of the text.
Methods/Approach:
Authors test for the impact of specific dual pathologies, and substance abuse type (eg, opioid, cocaine) on likelihood of early treatment cessation; however, I do not see any test for interactions between these. Is it possible that dual pathology also predicts the type of substance abused? EG: Do individuals with personality disorders preferentially abuse a certain type of substance? It would be edifying to include some investigation of this, and to include interaction tests here. This may also go some way to elucidating why some dual pathologies lead to increased likelihood of dropping out of treatment.
Following the reviewer's suggestion, we have proceeded to analyze the association between dependence on different drugs and other comorbid mental disorders. The relationships found have been reported in the text.
“The analysis regarding dual pathology and drug use disorders showed that among the patients with psychopathological disorders, there was a statistically significant association with alcohol-related disorders (50.0% of those with dual pathology showed alcohol dependence compared with 40.2% patients without dual psychopathology, 2 = 15.89; p <. 001). Specifically, patients with alcohol dependence showed higher rates of mood disorders (8.6% vs. 3.8%; χ2 = 23.451; p < .001) and of anxiety, dissociative, stress-related, somatoform and other non-psychotic mental disorders (8.9% vs. 5.9%; χ2 = 8.165; p = .004). Patients with cannabis dependence showed higher rates of schizophrenia, schizotypal, delusional, and other non-mood psychotic disorders (10.1% vs. 3.3%; χ2 = 47.096; p < .001).
In contrast, we observed a lower prevalence of opioid dependence (18.8% of patients with psychopathology compared with 30.0% of patients without psychopathology, 2 = 23.42; p <. 001). Compared with patients without drug dependence, patients with dependence on this drug showed lower rates of mood disorders (4.2% vs. 7.2%; 2 = 7.194; p = .007) and anxiety, dissociative, stress-related, somatoform and other nonpsychotic mental disorders (8.4% vs. 5.2%; 2 = 7.136; p = .008). It was also observed that patients with cocaine dependence had lower rates of dual pathology. (48.7% of patients with psychopathology and 63.7% of patients without psychopathology, 2 = 36.26; p <. 001). Specifically, these patients presented lower rates of mood disorders (4.5% vs. 9.7%; 2 = 26.184; p < .001) along with anxiety, dissociative, stress-related, somatoform and other non-psychotic mental disorders (10.3% vs. 5.9%; 2 = 15.543; p < .001).
No association was observed between any type of drug dependence and harmful drug use, and personality disorders.”
Likewise, the interaction between dependence on different drugs and mental disorders has been analyzed in Cox regression analysis. No statistically significant coefficient value was found. This result has been reported in the text:
“Analysis of the interactions between dependence on different drugs and other comorbid mental disorders did not reveal any statistically significant coefficient.”
Although it is possibly outside the scope of this study, authors could consider a larger, hypothesis-free approach to probe a range of explanations for early treatment leaving; for example, would it be possible to perform a phenome-wide association study (PheWAS) using treatment outcome as the test variable? This may provide interesting additional insights beyond the analysis described here.
We thank the reviewer for this study proposal. The data of this study are a part of a database with more than 70,000 patients in treatment for drug abuse/dependence (inpatients and outpatients). Methodological approaches such as PheWAS may be interesting for data analysis.
Discussion:
Authors observe lower rates than expected of mood, anxiety and personality disorders within this sample. It seems likely to me that this may be due to the EHR-based approach; patients may have been diagnosed previously, at other centres, or have their diagnoses listed in clinician notes rather than in ICD codes. This limitation of using EHR should be discussed by authors.
Following the reviewer's suggestion, we have proceeded to incorporate this idea within the limitations of the study:
“However, the prevalence rates of comorbid mental disorders were found to be lower lower than those observed in other studies. It is worth questioning, therefore, whether recording diagnoses through the patients' clinical history could lead to an underestimation of the diagnoses of comorbid mental disorders.”
It would also be interesting to discuss the potential reasons that underly the relationship between these factors (substance abuse and dual pathologies) and treatment outcome in more detail.
Following the reviewer's recommendations, we have proceeded to expand the discussion to include this issue.
“In addition to the above, it should be borne in mind that patients with dual pathology usually present greater severity and complexity of symptoms. Therefore, the therapeutic approach to these patients requires professionals specialized in dual pathology (Tirado-Muñoz et al., 2018), who can plan a comprehensive intervention for mental health problems adapted to the needs of each patient (De Ruysscher et al., 2017). However, addiction centers do not always have the necessary resources for the treatment of these patients (Carrá et al., 2015; McGovern et al., 2014), which can generate dissatisfaction with the therapy and increase the risk of drop-out (Schulter, Meier & Stirling, 2011).”

Round 2
Reviewer 1 Report
This is a re-review.
Based on a sample of 2,437 inpatients treated for substance abuse, the authors examined the predictive value of co-morbid cluster B personality disorders. Patients were referred from outpatient centers when they either did not respond to therapy, or needed more intensive medical treatment. The vast majority of patients were male (84.2%). According to ICD10, half of the patients (53.4%) had received more than one diagnosis of substance dependence. The data analyses relied exclusively on information that the authors extracted from the electronic patient records. Outcome criteria were response to treatment, premature drop-out, and treatment retention.
The authors found a responder rate of 41.8% along with a higher drop-out risk for patients with cluster B personality disorders (p=0.009). And the authors concluded “specific interventions might be needed for this patient group in therapeutic communities”.
Comments
The authors responded quite well to the questions raised by the reviewers and have made key points of the manuscript much clearer.
The missing psychiatric diagnoses have been included, and the authors have detailed their findings with respect to these diagnoses, thus considerably increasing the scientific value of the manuscript.